# Oncogenic BRAF and p53 Interplay in Melanoma Cells and the Effects of the HDAC Inhibitor ITF2357 (Givinostat)

**DOI:** 10.3390/ijms24119148

**Published:** 2023-05-23

**Authors:** Adriana Celesia, Marzia Franzò, Diana Di Liberto, Marianna Lauricella, Daniela Carlisi, Antonella D’Anneo, Antonietta Notaro, Mario Allegra, Michela Giuliano, Sonia Emanuele

**Affiliations:** 1Department of Biomedicine, Neurosciences and Advanced Diagnostics (BIND), Biochemistry Building, University of Palermo, 90127 Palermo, Italy; adriana.celesia@unipa.it (A.C.); marzia.franzo@unipa.it (M.F.); diana.diliberto@unipa.it (D.D.L.); marianna.lauricella@unipa.it (M.L.); daniela.carlisi@unipa.it (D.C.); 2Laboratory of Biochemistry, Department of Biological, Chemical and Pharmaceutical Sciences and Technologies (STEBICEF), University of Palermo, 90127 Palermo, Italy; antonella.danneo@unipa.it (A.D.); antonietta.notaro@unipa.it (A.N.); mario.allegra@unipa.it (M.A.)

**Keywords:** HDAC inhibitor, ITF2357, BRAF, melanoma, p53, apoptosis

## Abstract

Oncogenic BRAF mutations have been widely described in melanomas and promote tumour progression and chemoresistance. We previously provided evidence that the HDAC inhibitor ITF2357 (Givinostat) targets oncogenic BRAF in SK-MEL-28 and A375 melanoma cells. Here, we show that oncogenic BRAF localises to the nucleus of these cells, and the compound decreases BRAF levels in both the nuclear and cytosolic compartments. Although mutations in the tumour suppressor *p53* gene are not equally frequent in melanomas compared to BRAF, the functional impairment of the p53 pathway may also contribute to melanoma development and aggressiveness. To understand whether oncogenic BRAF and p53 may cooperate, a possible interplay was considered in the two cell lines displaying a different p53 status, being p53 mutated into an oncogenic form in SK-MEL-28 and wild-type in A375 cells. Immunoprecipitation revealed that BRAF seems to preferentially interact with oncogenic p53. Interestingly, ITF2357 not only reduced BRAF levels but also oncogenic p53 levels in SK-MEL-28 cells. ITF2357 also targeted BRAF in A375 cells but not wild-type p53, which increased, most likely favouring apoptosis. Silencing experiments confirmed that the response to ITF2357 in BRAF-mutated cells depends on p53 status, thus providing a rationale for melanoma-targeted therapy.

## 1. Introduction

Epigenetics has been widely recognised as an important contributor to cancer transformation. The development of epi-drugs, including histone deacetylase inhibitors (HDACIs), has represented a useful tool to specifically target tumour cells displaying aberrant epigenetic profiles. Histone deacetylases (HDACs) often result in being overexpressed in tumours [1,2,3], causing chromatin hypoacetylation and repression of the genes involved in cell cycle arrest and apoptosis [4]. It has been widely shown that HDACIs reactivate these genes, thereby exerting a selective antitumour action [5,6,7] and representing promising epi-drugs in cancer therapy. ITF2357 (Givinostat) is a pan-HDAC inhibitor that shows remarkable antitumour potential either alone [8,9,10] or in combination with conventional chemotherapeutic drugs [11]. This compound is currently under clinical investigations for leukaemia and lymphomas [12,13].

We have recently provided evidence that ITF2357 displays antitumour efficacy in melanoma cells bearing the oncogenic BRAF^V600E^ mutation, as it is capable of specifically targeting this oncogene and promoting apoptosis [14].

The status of the proto-oncogene BRAF, a serine/threonine kinase of the mitogenic MAPK signalling pathway, profoundly influences the development and progression of cutaneous melanomas [15,16,17]. In particular, oncogenic *BRAF* mutations, including the most frequent, *BRAF*^V600E^, have been widely described in melanomas and are associated with poor prognosis [18,19,20].

The tumour suppressor p53 represents a key factor and a crucial interactor in molecular networks modulating various processes involved in cell fate, including cell cycle control and DNA repair and balance between cell survival and cell death [21,22]. The loss of function of p53 has been widely recognised as a tumour-promoting event, as it clearly implies impairment of its tumour suppressor and pro-apoptotic actions [23,24,25]. In some cases, a gain of toxic function transforms p53 into an oncogene, underlying its dual function in carcinogenesis and tumour progression [26]. While *p53* genetic mutations represent a common feature of diverse tumour types [27], the mutational alteration of p53 in melanomas is considered a nonfrequent event [28]. However, its content and function often appear abnormal in melanomas, and p53 knockdown has been shown to result in melanoma-decreased proliferation [29]. The p53 loss of function in melanomas has been related to reduced cytokine expression, reduced migration and increased sensitivity to BRAF inhibition [30], thus supporting the role of aberrantly accumulated p53 in melanoma malignancy. The impairment of the p53 pathway in melanoma has been also attributed to either the deregulation of mouse double minute 2 (MDM2), the major ubiquitin ligase involved in p53 degradation or inactivation of the cyclin-dependent kinase inhibitor 2A (CDKN2A) locus, encoding p16INK4A and p14ARF tumour suppressors [28]. Moreover, evidence has been provided that the altered expression of p53 family isoforms impacts melanoma aggressiveness [31]. Therefore, targeting p53 in melanomas has been recognised as a valid strategy [28,32].

Although both the alterations of BRAF and p53 signalling are determinant in melanomas, a direct interplay between the oncogenic forms of these proteins has not been characterised yet.

Different HDACIs, including the short chain fatty acid butyrate and hydroxamic acid SAHA (vorinostat), have been shown to promote wild-type p53-dependent apoptosis [33] and to sensitise BRAF^V600E^-mutated tumour cells to the effects of BRAF inhibitors [34,35].

In our previous paper, we showed that ITF2357 is much more efficacious than SAHA in BRAF^V600E^-mutated melanoma SK-MEL-28 and A375 cells [14]. These cell lines, which differ in p53 status, displayed a different sensitivity to the effects of ITF2357, with A375 (expressing wild-type p53) more susceptible than SK-MEL-28 cells. SK-MEL-28 cells display mutated p53. Specifically, the L145R that was defined “likely oncogenic” [36] and the R273H considered responsible for the “gain of oncogenic function” [37] were described in these cells.

This paper aims to evaluate whether the different p53 status in the two melanoma cell lines influences the response of ITF2357 and to investigate a possible interplay between oncogenic BRAF and p53. These elucidations may provide a rationale and a molecular basis for the possible use of ITF2357 in melanoma-targeted therapy.

## 2. Results and Discussion

### 2.1. The Apoptotic Effect of IT2357 and BRAF Localisation in SK-MEL-28 and A375 Melanoma Cells

In our previous paper, we provided evidence that the HDAC inhibitor ITF2357 targets oncogenic BRAF in melanoma cells and promotes a switch from prosurvival autophagy to apoptotic cell death [14]. Activation of a canonical apoptotic pathway in melanoma cells by ITF2357 was evidenced using caspase activation and poly-ADP-ribose polymerase (PARP) degradation, as described in the same paper. Moreover, a different susceptibility of BRAF-mutated SK-MEL-28 and A375 cells to the effect of the compound was evidenced, with the IC50 estimated at 4.2 μM for SK-MEL-28 and 1.7 μM for A375 cells at 48 h of treatment. In this paper, we confirm that A375 cells appear more susceptible to the effects of ITF2357 than SK-MEL-28 cells. Specifically, the dose- and time-dependent effects of ITF2357 on the viability of the two cell lines are reported in Figure 1. These results clearly showed the different sensitivity of the two cell lines and indicated that the compound was effective after a lag phase of about 16/24 h.

In addition, using concentrations near the IC50 values, a flow cytometry analysis following double staining annexin V/PI confirmed apoptosis activation, which occurred at 48 h with 5 μM in SK-MEL-28 cells and with 1.5 μM ITF2357 in A375 cells (Figure 2). Negligible apoptotic effects were observed at the early time point (16 h) in accordance with the cell viability evaluations.

Considering the oncogenic BRAF-targeting effect of the compound, in this paper, we specifically investigated oncogenic BRAF localisation. As reported in Figure 3, Western blot analysis revealed that oncogenic BRAF was present not only in the cytosolic fraction but also in the nuclear fraction of both melanoma cell lines. We also confirmed BRAF nuclear localisation using immunofluorescence (Figure 4) that displayed clear green stained nuclei in both cell lines, which were also highlighted by merging with Hoechst nuclei staining. These findings are in line with the observations of Abd Elmageed et al. who first found BRAF in the nucleus of melanoma cells [38]. BRAF is a component of MAP kinase mitogenic signalling, and its translocation to the nucleus may account for its oncogenic potential. In this regard, it is interesting to note that nuclear BRAF^V600E^ has been shown to promote aggressive behaviour and vemurafenib-resistance in thyroid cancer [39]. Notably, as shown in the same Figure 3 and Figure 4, active ITF2357 concentrations (10 μM for SK-MEL-28 and 2 μM for A375) were capable of reducing the levels of oncogenic BRAF in both compartments of SK-MEL-28 and A375 cells at 24 h. This time was properly chosen to avoid nuclear fragmentation due to full blown apoptosis, as these concentrations were high enough to stimulate the BRAF-decreasing effect without provoking remarkable effects on cell viability. Our previous data indicate that oncogenic BRAF almost completely disappeared in both cell lines after 48 h of treatment with the HDAC inhibitor, and this effect was shown to be mainly due to degradative events, including proteasome-mediated processes [14,40]. We cannot exclude that these events occur at the nuclear level as well, especially considering that nuclear proteasomes account for the degradation of proteins localised in the nucleus [41].

### 2.2. ITF2357 Differently Affects p53 in SK-MEL-28 and A375 Melanoma Cells

The nuclear localisation of oncogenic BRAF led us to hypothesise that it may influence transcriptional factors responsible for melanoma oncogenic behaviour. The tumour suppressor p53 status is different in the two melanoma cell lines that we analysed. Specifically, SK-MEL-28 cells display two p53 mutations that are classified as likely oncogenic [36] and gain of oncogenic function [37], whereas A375 cells display wild-type p53. Therefore, we hypothesised that the different susceptibility of the two cell lines to the effects of ITF2357 could, in part, depend on p53 status and that p53 might also interact with oncogenic BRAF at the nuclear level. To test these hypotheses, we first evaluated the effects of the compound on the p53 levels in the two cell lines. Western blot analysis was performed at 16 and 48 h, which were considered proper treatment times to detect the protein trend at precocious and late phases of ITF2357 stimulation. The concentrations of the compound were chosen in the two cell lines according to cell viability profiles and in line with the experiments referred to in BRAF nuclear localisation. As shown in Figure 5, data revealed that ITF2357 dramatically reduced the level of oncogenic p53 in SK-MEL-28 cells at 16 h, with the band almost disappearing, whereas it increased the wild-type p53 level in A375 at the same treatment time. Prolonging treatment to 48 h also induced a p53 decrease in these cells, most likely due to the degradative events correlated with apoptosis.

According to these results, it is possible to interpret that the oncogenic p53 form present in SK-MEL-28 cells is somehow targeted by ITF2357, while the wild-type p53 present in A375 cells precociously increases to promote the apoptotic response and is then degraded when apoptosis is completed. To elucidate the dramatic p53 decrease in SK-MEL-28 cells, experiments were carried out in the presence of the proteasome inhibitor bortezomib and the autophagy inhibitor bafilomycin A1. To avoid possible synergistic interaction between bortezomib with the HDAC inhibitor, the concentration of ITF2357 was reduced to 5 μM in these experiments, and a short treatment time (16 h) was considered. As shown in Figure 6A, bortezomib, but not bafilomycin A1, was capable of completely preventing the effect of ITF2357 on p53. These data, together with the observation that ITF2357 increased the phosphorylated and active form of MDM2, the p53 E3 ligase (Figure 4B), suggested that the compound rapidly induces proteasome-mediated degradation of oncogenic p53 (o-p53). It remains to be elucidated whether o-p53 can be preferentially subjected to proteolysis compared to the wild-type one.

It is possible to speculate that the two p53 mutations present in SK-MEL-28 render the protein easily targetable under the effects of the HDAC inhibitor. In line with this hypothesis, data present in the literature sustain that selective degradation of mutated o-p53 may occur and can represent a strategy to target tumour cells [42,43,44]. On the other hand, wild-type p53 stabilization represents an alternative approach to favour cell cycle arrest and apoptosis counteracting tumour progression and invasiveness [45,46].

### 2.3. Oncogenic BRAF Interacts with Oncogenic p53 in Melanoma Cells

To understand whether nuclear BRAF interacts with p53, immunoprecipitation experiments were performed in the two melanoma cell lines. Data reported in Figure 7 clearly indicate that o-p53 is present in BRAF immunoprecipitates obtained using SK-MEL-28 cells (left panel). As a confirmation, the immunoprecipitation of p53 revealed the presence of BRAF (right panel). It interesting to note that ITF2357 targeted o-p53 in line with the previous Western blot analysis and reduced its interaction with BRAF.

The same evaluations in A375 cells displaying wild-type p53 did not produce an equally clear result. p53 was almost undetectable in the BRAF immunoprecipitates (left panel), and only a weak BRAF band was visible in the p53 immunoprecipitates (right panel), which completely disappeared after treatment with ITF2357. Ongoing studies in our laboratory aim to clarify whether the wild-type p53 form may interact with BRAF. Nevertheless, to our knowledge, no data are present in the literature in this regard.

Considering the immunoprecipitation results, it is possible to deduce that BRAF might preferentially bind to o-p53 and probably contribute to its oncogenic potential. To confirm that this interaction occurs at the nuclear level, we evaluated the localisation of o-p53 in SK-MEL-28. As shown in Figure 8, o-p53 appeared almost entirely localised in the nucleus, thereby suggesting that the interplay with BRAF occurs at this level. Again, the effect of ITF2357 was the dramatic reduction of o-p53, which was confirmed in this compartment.

To our knowledge, these results represent the first evidence of a possible BRAF^V600E^/o-p53 interplay in melanoma cells, and poor information is available in the literature in this regard. However, different papers indicate that both BRAF and p53 represent predictable factors in melanoma development and progression [29,47,48]. Previous observations by Patton et al. indicated that BRAF^V600E^ promoted nevi formation that rapidly developed into invasive melanomas in a p53-deficient zebrafish model [49].

According to this paper and other observations [50], p53 status may be determinant in melanoma onset and pharmacological responses. The canonical p53 tumour suppressor function results compromised in melanoma by different mechanisms [28]. Moreover, it is widely recognised that mutations changing p53 from a tumour suppressor to an oncogene strongly contribute to tumour transformation [51,52]. Particularly interesting are those p53 isoforms with enhanced expression and exerting a role in acquired resistance to BRAF inhibitors in melanoma cells [53], suggesting that targeting their functions may represent a valid approach to overcoming resistance to BRAF inhibitors in melanoma. The interaction of p53 with other components of the mitogenic MAPK/ERK pathway has been described. For instance, Wang et al. recently showed that the ERK2-p53 complex diverts p53 from its canonical tumour suppressor role and that the MEKK inhibitor trametinib promotes dissociation of this complex-restoring p53 function, promoting apoptosis in stomach/colorectal tumours [54]. Differently, our results suggest an interplay between an oncogenic form of p53 with oncogenic BRAF that may account for melanoma malignancy, supporting the efficacy of ITF2357 in targeting both oncogenic factors and disrupting their interaction.

### 2.4. The ITF2357 Response in Melanoma Cells Is Dependent on p53 Status

In line with the above considerations on p53, we considered relevant to verify whether p53 status may influence the response to ITF2357 in the two melanoma cell lines. To this purpose, p53 silencing was performed using RNA interference in both SK-MEL-28 and A375 cells. The Western blot results shown in Figure 9A,C confirmed the efficacy of the silencing. Interestingly, an evaluation of cell viability revealed that oncogenic p53 knockdown increased the effect of ITF2357 in SK-MEL-28 compared to the scramble siRNA control. In fact, cell viability in ITF2357-treated cells for 24 h decreased from 86.8% in scr-siRNA-transfected cells to 58.7% in p53 siRNA-transfected cells (Figure 9B). On the other hand, the knockdown of wild-type p53 in A375 cells prevented the effect of ITF2357. In this case, cell viability increased from 65.8% in scr-siRNA-transfected cells to 85% in p53 siRNA-transfected cells (Figure 9D). These data corroborate the oncogenic role of p53 in SK-MEL-28 cells, where it probably serves a prosurvival function, while in A375 cells, wild-type p53 maintains its usual tumour suppressor role promoting apoptosis.

These data also provide evidence that the ITF2357 response in BRAF-mutated melanoma cells is dependent on p53 status, thus providing a rationale for melanoma-targeted therapy. It has been widely documented that HDAC inhibitors promote p53-dependent apoptosis in different tumour models [55,56]. The presence of wild-type p53 in melanomas may, therefore, represent a predictor and a good condition to consider HDAC inhibitors as antitumour agents. ITF2357 has been shown to display a potent effect in melanoma cells compared to SAHA (vorinostat) [14], and its particular efficacy in (wild-type p53) A375 cells encourages its potential use for those melanoma types expressing functional p53. Considering that mutations of the p53 gene represent one of the most common genetic alterations in human cancer, targeting mutated (oncogenic) p53 constitutes another key approach for melanomas bearing p53 mutations. Our finding that ITF2357 targets both oncogenic p53 and oncogenic BRAF, disrupting their interaction in SK-MEL-28 cells, lays the foundations to consider this compound as a potential epigenetic candidate for melanoma-targeted therapy.

## 3. Materials and Methods

### 3.1. Chemicals and Reagents

ITF2357 (Givinostat) was synthesised and kindly provided by Italfarmaco, Cinisello Balsamo, MI, Italy. For the experiments, ITF2357 was dissolved in DMSO (20 mM stock solution) and stored at −20 °C. Final concentrations were then realised in culture medium. Equal volumes of DMSO were added to untreated cells as vehicle control. The autophagy inhibitor bafilomycin A1 and the proteasome inhibitor bortezomib were purchased from Sigma-Aldrich (Milan, Italy). Prior to use, all different stock solutions were opportunely diluted in DMEM culture medium to realise the proper final concentrations, not exceeding 0.01% (*v*/*v*) DMSO.

### 3.2. Cell Cultures

Human melanoma SK-MEL-28 (American Type Culture Collection, ATCC, Milan, Italy) and A375 (kindly provided by Prof. Luisa Tesoriere, University of Palermo, Palermo, Italy) cell lines were grown in monolayer in 75 cm^2^ flasks in DMEM medium, supplemented with 10% (*v*/*v*) heat-inactivated fetal bovine serum (FBS), 2 mM L-glutamine, 100 U/mL penicillin, and 50 µg/mL streptomycin in a humidified atmosphere of 5% CO_2_ in air at 37 °C. For the experiments, cells were seeded at a density of 1.5 × 10^5^/well (SK-MEL-28) or 1.8 × 10^5^/well (A375) in 6-well plates, respectively, and allowed to adhere overnight. Subsequently, cells were treated with 5 or 10 μM (SK-MEL28) and 1.5 or 2 μM (A375), and the incubation was protracted for the established times according to the specific experimental design. Materials and reagents for cell cultures were purchased from Euroclone (Pero, Italy) and Life Technologies Ltd. (Monza, Italy).

### 3.3. Annexin V-FITC/PI Staining

For these evaluations, Annexin V-FITC kit (cat n. 130-092-052) from Miltenyi Biotec (Bergisch Gladbach, Germany) was used. In detail, SK-MEL-28 (1.5 × 10^5^ cells/well) and A375 (1.8 × 10^5^ cells/well) were seeded in 6-well plates and, after 24 h, were treated with 5 µM (SK-MEL-28) or 1.5 µM (A375) ITF2357 for 48 h. Then, cells were harvested, washed twice in PBS, counted, and incubated (10^6^ cells) for 15 min with 5 µL annexin V/PI in a 100 µL binding buffer. The samples were then diluted with 500 µL binding buffer and analysed using flow cytometry with FACSCanto cytometer (BD Biosciences Company, Franklin Lakes, NJ, USA). Data were analysed using FlowJo v10 software (BD Biosciences).

### 3.4. Extraction of Cytosolic and Nuclear Fraction

SK-MEL-28 and A375 cells were cultured in 75 cm^2^ flasks upon reaching about 70% confluence. Next, 10 μM (SK-MEL-28) or 2 µM (A375) ITF2357 was then added and maintained for 24 h. Cells were then washed in PBS and scraped with 500–700 μL (depending on the cell line) of subcellular fractionation buffer (250 mM sucrose, 20 mM HEPES, 10 mM KCl, 1.5 mM MgCl_2_, 1 mM EDTA, 1 mM EGTA, 1 mM DTT, and protease inhibitors, pH 7.4), as previously described [57]. Subsequently, cells were passed 10 times through a needle of 25 G and kept on ice for 20 min. The homogenates were centrifuged at 720× *g* for 5 min at 4 °C. The pellets were resuspended in lysis buffer and passed 10 times through a needle of 25 G and centrifuged again at 720× *g* for 10 min at 4 °C. The pellets of the second centrifugation (nuclear fraction) were lysed with nuclear buffer (standard lysis buffer with 10% glycerol and 0.1% SDS–1% NP-40, 0.5% sodium deoxycholate, 0.1% SDS, and inhibitors of proteases, namely 25 µg/mL aprotinin, 1 mM PMSF, 25 µg/mL leupeptin and 0.2 mM sodium pyrophosphate) and sonicated. The supernatants obtained from the first centrifugation were considered as cytosolic fractions. Nuclear and cytosolic fractions were used to evaluate BRAF using Western Blot analysis. β-tubulin and histone H3 were used as cytoplasmic and nuclear markers, respectively.

### 3.5. Immunofluorescence Analysis

To confirm nuclear BRAF localisation, 1.5 × 10^5^ (SK-MEL-28) or 1.8 × 10^5^ (A375) cells were seeded in 24-wells plates and treated with ITF2357 for 24 h. Subsequently, cells were washed three times with PBS and fixed using 4% paraformaldehyde for 10 min. Cells fixed were then washed three times with PBS and treated with 0.1% Triton X-100 in PBS for 5 min. Incubation with mouse monoclonal antihuman BRAF (Santa Cruz Biotechnology) antibody diluted 1:200 in PBS containing 1% BSA and 0.05% sodium azide was protracted overnight at 4 °C. At the end of incubation, cells were washed three times with PBS and incubated with secondary antibody antimouse FITC (1:250 PBS with 1% BSA and 0.05 sodium azide) for 2 h at room temperature. Cells were then stained with 1 µg/mL Hoechst 33342 (Invitrogen; Thermo Fisher Scientific, Inc. Eugene, OR) for 10 min in dark. After PBS washings, cells were analysed under a OPTIKA IM3FL4 fluorescence microscope equipped with a digital imaging camera system (OPTIKA S.r.l, Ponteranica (BG), Italy) using Dapi (excitation wavelength of 355 nm and emission wavelength of 435 nm) and FITC (excitation wavelength of 490 nm and emission wavelength of 520 nm) filters. All images were taken at magnification of 200 and 400X. Images are representative of two independent experiments.

### 3.6. Immunoprecipitation Analysis

Immunoprecipitation (IP) was used to identify molecules that interact with specific proteins using a target protein-specific antibody in conjunction with protein A/G affinity beads. Co-IP is one of the standard methods of identifying or confirming the occurrence of protein–protein interaction events in vivo [58]. Briefly, for these evaluations SK-MEL-28 and A375 cells were cultured in 75 cm^2^ flasks and treated with 10 μM (SK-MEL-28) and 2 μM (A375) ITF2357 for 24 h. Cells were then washed in PBS and lysed with modified RIPA buffer (50 mM HEPES, 150 mM NaCl, 1% TRITON, 10% glycerol, 1,5 mM MgCl_2_ and 1 mM EGTA and protease inhibitors). Cells were thus scraped, and the lysates were collected and incubated on ice for 20 min. The lysates were passed 10 times through a needle of 25 G, and the homogenates were centrifuged at 13,000× *g* for 10 min. Afterwards, the supernatants were used for protein assay, and equal amounts of proteins (600 μg) were incubated with mouse monoclonal BRAF (Santa Cruz: sc-5284) or p53 DO1 (Santa Cruz: sc-126) antibody for 2 h on ice (ratio: 1 μg antibody/1 mg proteins). Then, 30 μL of A/G agarose beads (Santa Cruz 5284) were added, and the incubation was protracted overnight at 4 °C. Immunocomplexes were then centrifuged at 10,000× *g* for 5 min, and the beads were washed three times in PBS. The immunocomplexes were then dissolved in 2X SDS loading buffer at 90 °C for 5 min, and the samples were centrifuged to eliminate the beads. Proteins present in the immunocomplexes were then analysed using Western Blot analysis with different antibodies (BRAF or p53-DO1—Santa Cruz Biotechnology, DBA, Milan, Italy).

### 3.7. P53 siRNA Transfection

For these evaluations, 1.5 × 10^5^ (SK-MEL-28) or 1.8 × 10^5^ (A375) cells were seeded in 6-wells plates. Once about 70% confluence was reached, 50 pmoles of siRNA (p53 siRNA (h2), Santa Cruz Biotechnology, Dallas, USA) were transfected in 1 mL DMEM containing 1% glutamine without serum and antibiotics, as previously described [59]. Before transfection, siRNAs were incubated with the transfection reagent Lipofectamine™ 2000, (Invitrogen™) (5 µL/well) for 30 min at room temperature. Then, cells were transfected for 6 h with p53 siRNA or with siRNA scramble (60 nM, SI03650318, Qiagen, Hilden, Germany). Afterwards, complete medium was added to each well, and cells were treated for further 24 with ITF2357. To confirm the silencing of the target protein, Western Blot analysis of p53 was performed.

### 3.8. Evaluation of Cell Viability

SK-MEL-28 and A375 cell viability was determined using 3-(4,5-dimethylthiazol-2-yl)- 2,5-diphenyltetrazolium bromide (MTT) assay, as previously described [60]. Briefly, after treatment in 96-well plates (triplicate wells for each sample), MTT solution (final concentration 5.5 mg/mL) was added for 2 h. The medium was then replaced with 100 μL lysis buffer (20% sodium dodecyl sulfate in 50% N,N- dimethylformamide), and the absorbance of the formazan product released was measured at 490 nm with 630 nm as a reference wavelength using an automatic ELISA plate reader (OPSYS MR, Dynex Technologies, Chantilly, VA, USA). Values reported in Figures are expressed as percentage of the viability of treated cells compared with vehicle treated (untreated control, 100% viability).

### 3.9. Western Blot Analysis

For Western blot analysis, whole-cell extracts were prepared in ice-cold lysis RIPA buffer (1% NP-40, 0.5% sodium deoxycholate and 0.1% SDS in PBS; pH 7.4), supplemented with a protease inhibitor cocktail, and subjected to SDS PAGE and consequent immunoblot. In these experiments, the correct protein loading was verified using both Ponceau red staining and housekeeping protein γ-tubulin immunodetection, as previously described [61]. Specific primary antibodies directed against BRAF and p53-DO1 (diluted 1:500) and β-tubulin (diluted 1:1000) were purchased from Santa Cruz Biotechnology (St. Cruz, CA, USA); γ-tubulin and H3 histone (diluted 1:1000) were purchased from Sigma-Aldrich (Milan, Italy), while p-MDM2 (diluted 1:1000) was purchased from Cell Signaling Technology (Beverly, MA, USA). Immunodetection was carried out using electrochemical luminescence labelling system (ECL) with ChemiDoc, XR Image system (Bio-Rad Laboratories, Hercules, CA, USA). The intensity of the protein bands was quantified using Quantity One 4.6.6 1-D Imaging Software (Bio-Rad Laboratories) and reported as the ratio of the intensity of protein bands normalised to γ-tubulin versus the intensity of the untreated samples, if not differently indicated [62].

### 3.10. Statistical Analysis

Data were represented as mean ± S.D., and analyses were performed using the Student’s t-test and one-way analysis of variance. Comparisons between untreated control vs. all treated samples were made. If a significant difference was detected using ANOVA analysis, this was re-evaluated using post hoc Bonferroni’s test. GraphPad PrismTM 4.0 Software (GraphPad PrismTM Software Inc., San Diego, CA, USA) was used for statistical calculations. The statistical significance threshold was fixed at *p* < 0.05.

## 4. Conclusions

Overall, data presented in this paper provide evidence for the first time that oncogenic BRAF and p53 interact in melanoma cells. The HDAC inhibitor ITF2357 targets oncogenic BRAF that is localised in both the cytoplasm and the nucleus of melanoma cells. The compound exerts a different effect on p53 protein levels depending on the p53 status, as it is capable of targeting the oncogenic form while increasing the wild-type form. Accordingly, the HDAC inhibitor reduced oncogenic BRAF and p53 interaction. P53 silencing confirmed that the response to ITF2357 may also depend on the functional status of p53, whose presence could make melanoma more susceptible to HDACI treatment. However, the capability of ITF2357 in targeting oncogenic p53 as well as oncogenic BRAF renders the epi-drug a promising compound in melanoma-targeted therapy.

## Figures and Tables

**Figure 1 ijms-24-09148-f001:**
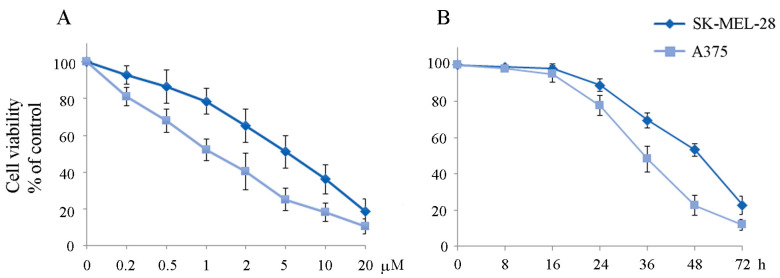
Dose- and time-dependent effects of ITF2357 on the viability of melanoma cells. SK-MEL-28 and A375 cells were treated for 48 h with the indicated concentrations of ITF2357 (**A**) or with 5 μM ITF2357 for 48 h (**B**). Cell viability was evaluated using MTT assay as reported in Materials and Methods. The histograms are representative of three independent experiments.

**Figure 2 ijms-24-09148-f002:**
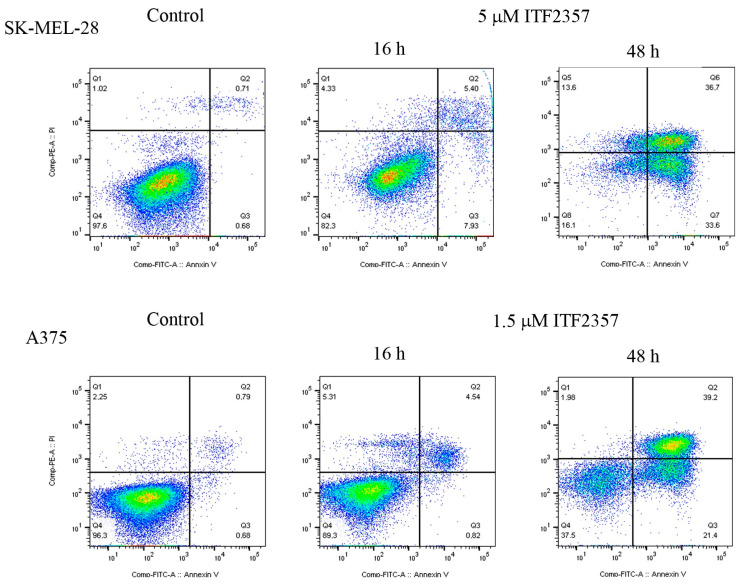
ITF2357 induces apoptosis in melanoma cells. SK-MEL-28 and A375 cells were treated with 5 μM and 1.5 μM ITF2357, respectively, for 16 or 48 h and subjected to annexin V apoptosis detection kit as reported in Materials and Methods. The figure reports untreated control evaluated at 48 h only, as the 16 h control produced similar results in both cell lines. Analysis was performed using flow cytometry with FACSCanto BD. The percentage of annexin V positive cells was evaluated using FlowJo BD v10 software (BD Biosciences). The results are representative of two independent experiments.

**Figure 3 ijms-24-09148-f003:**
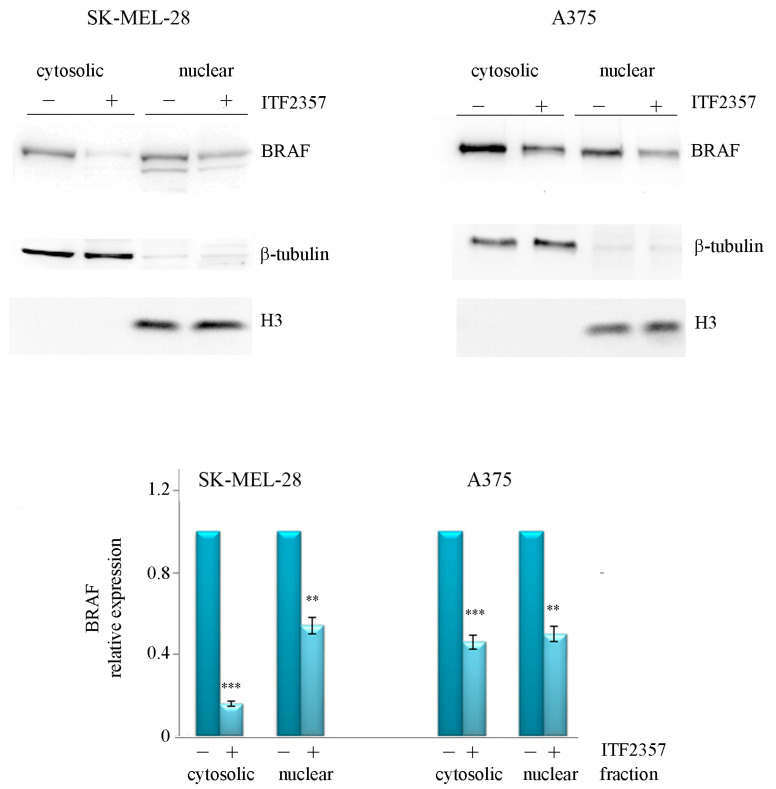
Nuclear localisation of oncogenic BRAF and the effects of ITF2357. SK-MEL-28 and A375 cells were treated with 10 μM and 2 μM ITF2357 for 24 h. Cytosolic and nuclear fractions were then prepared as reported in Materials and Methods and subjected to Western blot analysis. Representative blots of three independent experiments and densitometric analysis are shown, referred to as beta tubulin for cytosolic fractions and H3 histone for nuclear fractions. ** *p* < 0.01 and *** *p* < 0.001 with respect to untreated cells.

**Figure 4 ijms-24-09148-f004:**
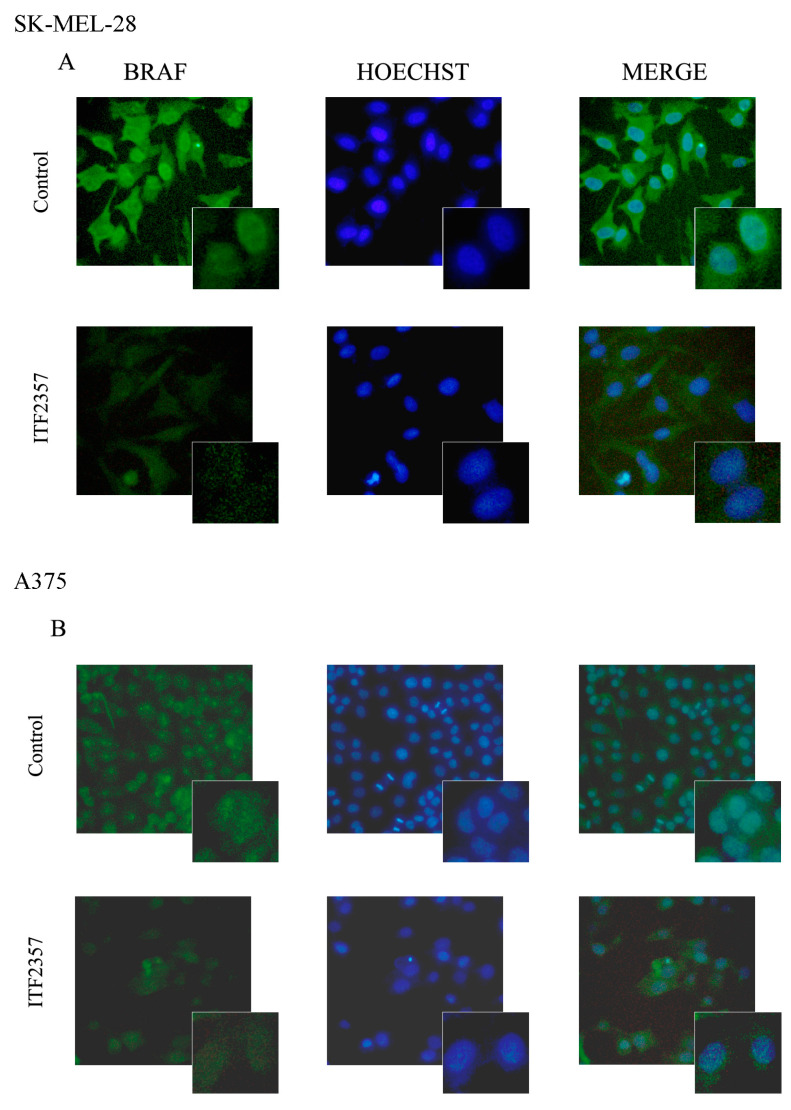
Visualization of BRAF nuclear localisation using immunofluorescence. SK-MEL-28 (**A**) and A375 (**B**) were treated for 24 h with 10 μM and 2 μM, respectively. Cells were then fixed with paraformaldehyde, permeabilised with Triton X-100, stained with mouse anti-BRAF antibody and subsequently with secondary antimouse antibody FITC-conjugated (green fluorescence). Nuclei were stained with Hoechst 33342 (blue fluorescence) as reported in Materials and Methods. Cells were analysed under a fluorescence microscope equipped with a digital imaging camera system using Dapi and FITC filters. All images were taken at magnification of 200× (big images) and 400× (small images). The images are representative of two independent experiments.

**Figure 5 ijms-24-09148-f005:**
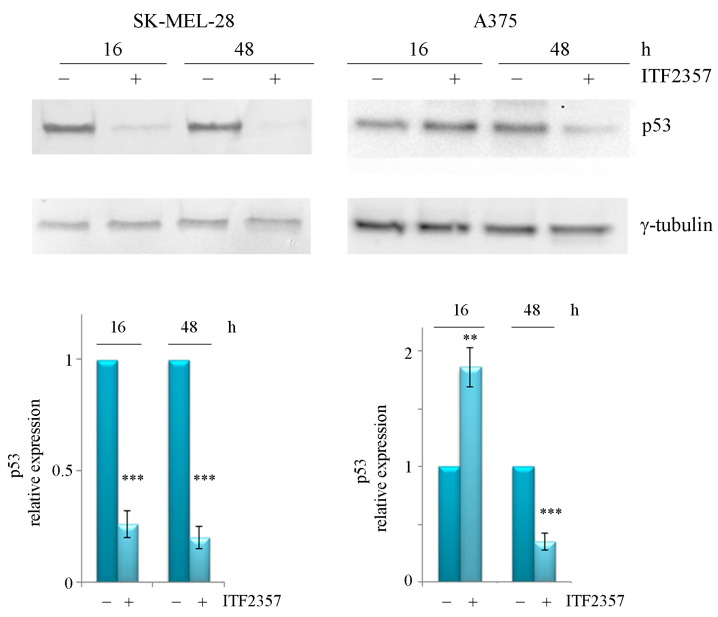
Different effects of ITF2357 on p53 protein levels. SK-MEL-28 and A375 cells were treated with 10 μM and 2 μM ITF2357, respectively, for the indicated times. Western blot analysis of p53 was performed as reported in Materials and Methods. Representative blots of two independent experiments and densitometric analysis are shown. ** *p* < 0.01 and *** *p* < 0.001 with respect to untreated cells.

**Figure 6 ijms-24-09148-f006:**
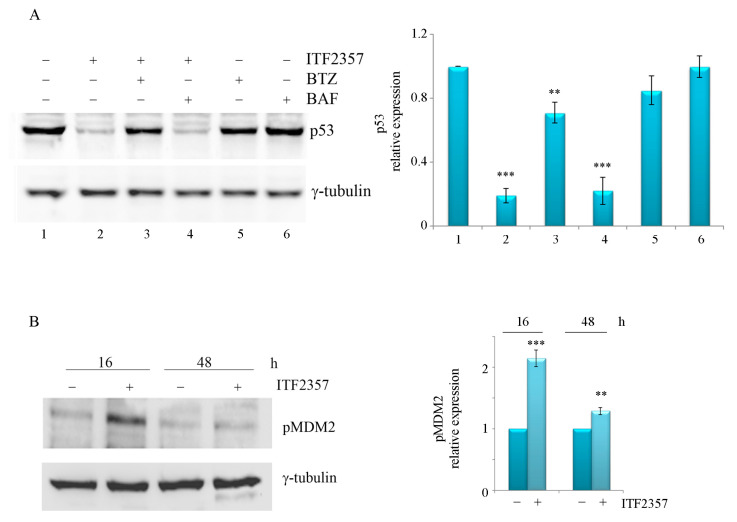
The effects of bortezomib and bafilomycin A1 on oncogenic p53 reduction and the effect of ITF2357 on pMDM2. (**A**) SK-MEL-28 cells were treated for 16 h with 5 μM ITF2357 in the presence of 15 nM bortezomib (BTZ) or 20 nM bafilomycin A1 (BAF). (**B**) SK-MEL-28 cells were treated for the indicated times with 10 μM ITF2357. Western blot analysis of p53 (**A**) and pMDM2 (**B**) was performed as reported in Materials and Methods. Representative blots of two independent experiments and densitometric analysis are shown. ** *p* < 0.01 and *** *p* < 0.001 with respect to untreated cells.

**Figure 7 ijms-24-09148-f007:**
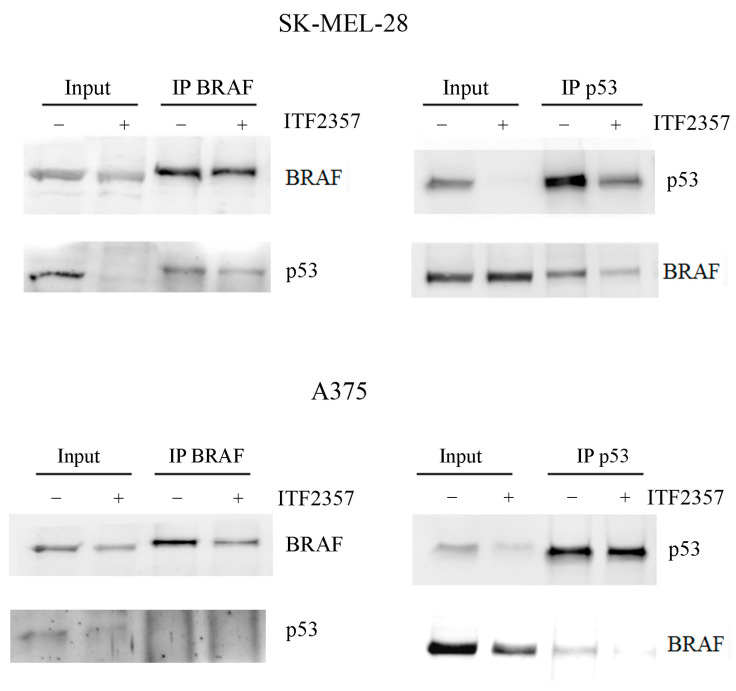
Association of oncogenic BRAF with oncogenic p53. SK-MEL-28 and A375 cells were treated with 10 μM and 2 μM ITF2357 for 24 h. Immunoprecipitation of BRAF (**left** panel) or p53 (**right** panel) and Western blot analysis were performed as reported in Materials and Methods. Representative blots of two independent experiments are shown.

**Figure 8 ijms-24-09148-f008:**
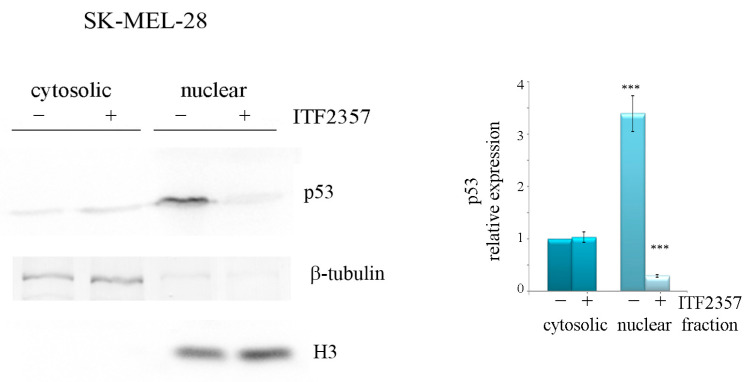
Nuclear localisation of oncogenic p53 and the effects of ITF2357. SK-MEL-28 cells were treated with 10 μM ITF2357 for 24 h. Cytosolic and nuclear fractions were then prepared as reported in Materials and Methods and subjected to Western blot analysis. Representative blots of two independent experiments and densitometric analysis are shown, referred to as beta tubulin for cytosolic fractions and H3 histone for nuclear fractions. *** *p* < 0.001 with respect to untreated cells.

**Figure 9 ijms-24-09148-f009:**
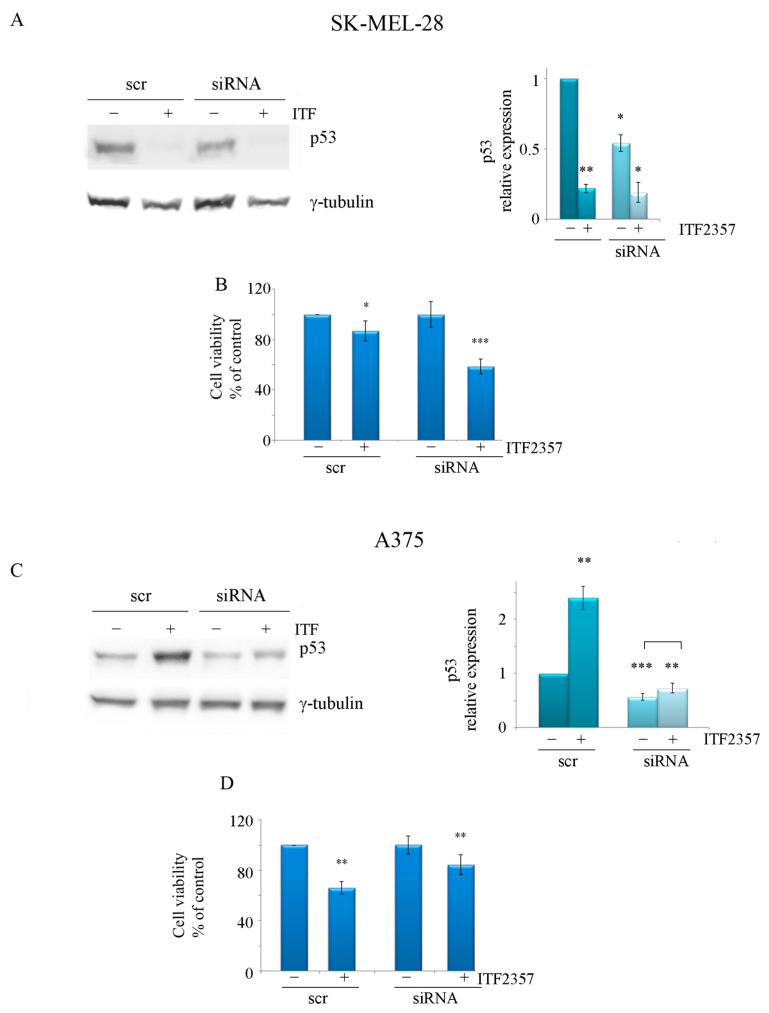
The effect of ITF2357 in p53 silenced cells. P53 silencing was performed in SK-MEL-28 and A375 cells as reported in Materials and Methods. Silenced cells were then treated with 5 μM (SK-MEL-28) and 2 μM (A375) ITF2357 for 24 h. Representative blots of two independent experiments and densitometric analysis are shown to confirm the reduction in p53 in silenced cells (**A**,**C**). Cell viability was then determined using MTT analysis to verify the effect of ITF2357 in silenced cells (**B**,**D**). Histograms are representative of two independent experiments. * *p* < 0.05, ** *p* < 0.01 and *** *p* < 0.001 with respect to the corresponding untreated cells.

## Data Availability

Not applicable.

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
