# Peer review of "Oncogenic BRAF and p53 Interplay in Melanoma Cells and the Effects of the HDAC Inhibitor ITF2357 (Givinostat)"

_ijms, 2023, doi:10.3390/ijms24119148_

Round 1

Reviewer 1 Report

The authors well describe the biochemical mechanism of the HDAC inhibitor ITF2357 in two melanoma cells. They highlight the key role of p53 and its interaction with BRAF. Additionally, knockdown of p53 caused different effect on cell viability of SK-MEL-28 and A375 demonstrating that p53 plays different role in cancer cells. Few comments need to addressed as follow:

1) Did the authors observe an interaction of BRAF and p53 in the nucleus of SK-MEL-28 cells?

2) Did the authors measure annexin at different time points? Any sign of early apoptosis before 48h ?

3) In A375 cells, authors used a lower concentration compared to that one used in SK-MEL. How was chosen that concentration?

Minor editing of English language required

Author Response

Reviewer 1 The authors well describe the biochemical mechanism of the HDAC inhibitor ITF2357 in two melanoma cells. They highlight the key role of p53 and its interaction with BRAF. Additionally, knockdown of p53 caused different effect on cell viability of SK-MEL-28 and A375 demonstrating that p53 plays different role in cancer cells. Few comments need to addressed as follow:

Many thanks for the positive evaluation of our manuscript by Reviewer 1. We appreciated the suggestions that definitely aim to improve the paper quality. Below our response point by point.

1) Did the authors observe an interaction of BRAF and p53 in the nucleus of SK-MEL-28 cells?

The nuclear BRAF localization in melanoma cells prompted us to hypothesize that interaction with oncogenic p53 occurs at this level. We thank the Reviewer for suggesting verifying this hypothesis.

To this aim, we performed new experiments to evaluated oncogenic p53 localization in SK-MEL-28 cells. The results indicated that oncogenic p53 is almost entirely localized in the nucleus and therefore interaction may occur in this compartment. Moreover, ITF2357 caused the disappearance of the band at this level, thus confirming the results reported in the paper indicating that the HADC inhibitor targets oncogenic p53 in SK-MEL-28. We provided to include new Figure 8 in the manuscript and a new sentence in this regard in the results and discussion section (lines 228-233).

2) Did the authors measure annexin at different time points? Any sign of early apoptosis before 48h?

To address this comment, we also included annexin V staining at early time point (16 h). However, according to cell viability evaluations reported in new Figure 1, the apoptotic effect of ITF2357 in melanoma cells occurs after a lag phase of about 24h and becomes evident at 48 h. Indeed, the percentage of annexin V positive cells at the early time point was negligible whereas full blown apoptosis occurred at 48 h. We provided to include the early time point in new Figure 2 as requested and we discussed this result (lines 111-112).

3) In A375 cells, authors used a lower concentration compared to that one used in SK-MEL. How was chosen that concentration?

The differences in the concentrations used for the two melanoma cell lines depend on their different sensitivity to the effect of the compound. Specifically, the manuscript showed the IC50 values at 48 h and we provided to include the dose- and time-dependent effects on cell viability to better highlight this difference (new Figure 1), and we included a sentence in this regard in the text (lines 96-99). The much higher sensitivity of A375 cells induced us to employ 2 mM ITF2357 in most experiments which is a concentration near to the IC50. On the other hand, we used 5 or 10 mM for SK-MEL-28 depending on the experiments, pushing to 10 mM for those experiments at early time points (16 or 24 h). 

Reviewer 2 Report

This article by Celesia et al. studied a regulatory mechanism between BRAF and P53 by a HDAC inhibitor in melanoma cells. To support the conclusions, questions below need to be answered.

1. Figure 2: Without knowing the total level change of BRAF, it's really hard to say there is a translocation. The quantification only shows the decrease of BRAF in response to the drug. An IF of BRAF and cytoplasmic or nuclear markers will help address the possible translocation.

2. Figure 3: Why was the drug used at different doses? The difference authors had observed could be from the dose.

3. Figure 5: Given the differential level in input, it's difficult to interpret the IP results.

4. Figure 6: The P53 knockdown using siRNA is very poor. Authors need to improve knockdown efficiency, which may alter the observation. Also distinct siRNAs should be used to validate findings.

Author Response

Reviewer 2 This article by Celesia et al. studied a regulatory mechanism between BRAF and P53 by a HDAC inhibitor in melanoma cells. To support the conclusions, questions below need to be answered.

We thank the Reviewer for the consideration of our manuscript and for the suggestions that allow us to clarify some points, as described in detail below.

  1. Figure 2: Without knowing the total level change of BRAF, it's really hard to say there is a translocation. The quantification only shows the decrease of BRAF in response to the drug. An IF of BRAF and cytoplasmic or nuclear markers will help address the possible translocation.

The total level change of BRAF was deeply described in our previous paper (Celesia et al. Biomedicines 17;10(8):1994, 2022) that was cited in this manuscript (ref 14). However, data reported in this manuscript indicate a basal level of BRAF in the nucleus, so rather than referring to nuclear translocation, we considered BRAF nuclear localization which is most likely related with its oncogenic potential as discussed in the manuscript (lines 127-131). As the Reviewer pointed out, the quantification shows the decrease of BRAF in response to the compound, which is the relevant result.  That means that ITF2357 is also capable of reducing nuclear BRAF level. As suggested by the Reviewer, to corroborate these results, we also performed IF experiments which have been included in new Figure 4 and discussed in the text (lines 123-126).  

  1. Figure 3: Why was the drug used at different doses? The difference authors had observed could be from the dose.

The differences in the concentrations used for the two melanoma cell lines depend on their different sensitivity to the effect of the compound. To better highlight this difference, we included dose- and time- dependent effects of ITF2357 on cell viability (new Figure 1). The much higher sensitivity of A375 cells induced us to employ 2 mM ITF2357 in most experiments which is a concentration near to the IC50. On the other hand, we used 5 or 10 mM for SK-MEL-28 (which are more resistant) depending on the experiments pushing to 10 mM for those experiments at early time points (16 or 24h). More specifically, the rationale choice of the concentrations used in the p53 evaluation experiment was based on the cell viability results indicating that 2 mM and 10 mM ITF2357 produced similar effects in A375 and SK-MEL-28 cells respectively.  Since the effect on p53 level was evaluated at both early (16 h) and late time point (48h) for western blot analysis, we considered proper the above-mentioned concentrations. Please note that these choices are indicated in the Materials and Methods section (lines 318-320) However, we provided to include a further explanation sentence in this regard also in the results and discussion section (lines 170-172).  

  1. Figure 5: Given the differential level in input, it's difficult to interpret the IP results.

In our opinion, the IP results clearly indicate interaction between BRAF and oncogenic p53. This was evident in SK-MEL-28 cells (bearing p53 oncogenic mutations) and was confirmed by reciprocal IP reported in Figure 5. We may agree about the difficult interpretation of data referred to A375 cells, but as we discussed in text, a poor interaction possibly occurs between the wild type p53 form and BRAF. We provided to include a new sentence in the text to better discuss this point (lines 219-221).

  1. Figure 6: The P53 knockdown using siRNA is very poor. Authors need to improve knockdown efficiency, which may alter the observation. Also distinct siRNAs should be used to validate findings.

The p53 knockdown efficiency was about 50% in both cell lines, which cannot be considered a poor result. Moreover, as reported in the Figure, the increase in p53 levels induced by the compound in A375 scramble cells was almost null in treated siRNA transfected cells thus confirming the silencing efficiency. The transfection reagent lipofectamine that we used for these experiments is one of the most performing. Unfortunately, we are not equipped with other facilities for different transfection methods to confirm the reported results, that by the way we consider clear. Moreover, the differential effects of the compound on cell viability in scramble and siRNA transfected cells confirmed the effective transfection and the different status of p53 in the two cell lines. The relevant finding was that silencing oncogenic p53 (SK-MEL-28) increased the effect of the compound, whereas silencing the wild type (A375) reduced the effect of the compound since wild type p53 realistically induces apoptosis.